# The Difficulty of Passive Learning in Deep Reinforcement Learning

**Georg Ostrovski**
DeepMind
ostrovski@deepmind.com

**Pablo Samuel Castro**
Google Research, Brain Team
psc@google.com

**Will Dabney**
DeepMind
wdabney@deepmind.com

## Abstract

Learning to act from observational data without active environmental interaction is a well-known challenge in Reinforcement Learning (RL). Recent approaches involve constraints on the learned policy or conservative updates, preventing strong deviations from the state-action distribution of the dataset. Although these methods are evaluated using non-linear function approximation, theoretical justifications are mostly limited to the tabular or linear cases. Given the impressive results of deep reinforcement learning, we argue for a need to more clearly understand the challenges in this setting. In the vein of Held & Hein's classic 1963 experiment, we propose the "tandem learning" experimental paradigm which facilitates our empirical analysis of the difficulties in offline reinforcement learning. We identify function approximation in conjunction with fixed data distributions as the strongest factors, thereby extending but also challenging hypotheses stated in past work. Our results provide relevant insights for offline deep reinforcement learning, while also shedding new light on phenomena observed in the online case of learning control.

## 1 Introduction

Learning to act in an environment purely from observational data (i.e. with no environment interaction), usually referred to as *offline reinforcement learning*, has great practical as well as theoretical importance (see [Levine et al., 2020] for a recent survey). In real-world settings like robotics and healthcare, it is motivated by the ambition to learn from existing datasets and the high cost of environment interaction. Its theoretical appeal is that stationarity of the data distribution allows for more straightforward convergence analysis of learning algorithms. Moreover, decoupling learning from data generation alleviates one of the major difficulties in the empirical analysis of common reinforcement learning agents, allowing the targeted study of learning dynamics in isolation from their effects on behavior.

Recent work has identified *extrapolation error* as a major challenge for offline (deep) reinforcement learning [Achiam et al., 2019, Buckman et al., 2021, Fujimoto et al., 2019b, Fakoor et al., 2021, Liu et al., 2020, Nair et al., 2020], with *bootstrapping* often highlighted as either the cause or an amplifier of the effect: The value of missing or under-represented state-action pairs in the dataset can be over-estimated, either transiently (due to insufficient training or data) or even asymptotically (due to modelling or dataset bias), resulting in a potentially severely under-performing acquired policy. The corrective feedback-loop [Kumar et al., 2020b], whereby value over-estimation is self-correcting via *exploitation* during interaction with the environment (while under-estimation is corrected by *exploration*), is critically missing in the offline setting.

35th Conference on Neural Information Processing Systems (NeurIPS 2021).

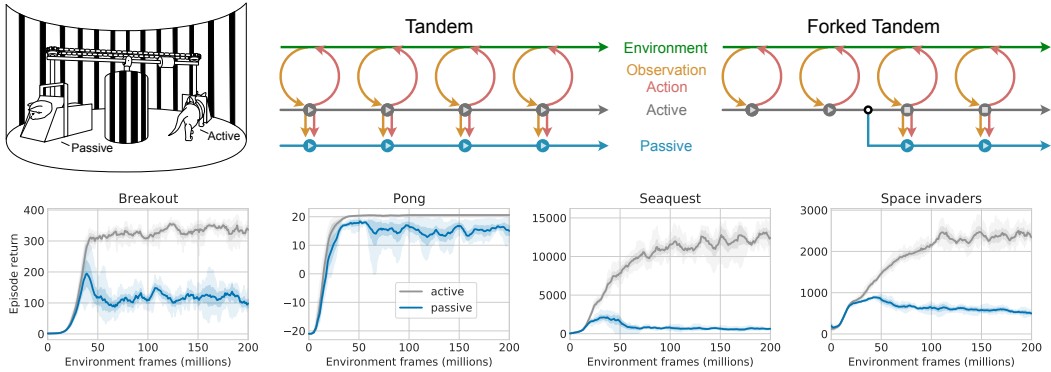

Figure 1: (**top-left**) Held and Hein [1963] experiment setup. (**top-right**) Illustrations of the Tandem and Forked Tandem experiment setups. (**bottom**) Tandem (active and passive) performance on 4 selected Atari domains. In all figures, *active* agent performance is shown in gray.

To mitigate this, typically one of a few related strategies are proposed: policy or learning update constraints preventing deviations from states and actions well-covered by the dataset or satisfying certain uncertainty bounds [Fujimoto et al., 2019a,b, Kumar et al., 2019, 2020c, Achiam et al., 2019, Wang et al., 2020b, Wu et al., 2021, Nair et al., 2020, Wu et al., 2019, Yu et al., 2020], pessimism bias to battle value over-estimation [Buckman et al., 2021, Kidambi et al., 2020], large and diverse datasets to improve state space coverage [Agarwal et al., 2020], or learned models to fill in gaps with synthesized data [Schrittwieser et al., 2021, Matsushima et al., 2020]. While many of these enjoy theoretical justification in the tabular or linear cases [Thomas et al., 2015], guarantees for the practically relevant non-linear case are mostly lacking.

In this paper we draw inspiration from the experimental paradigm introduced in the classic Held and Hein [1963] experiment in psychology. The experiment involved coupling two young animal subjects' movements and visual perceptions to ensure that both receive the same stream of visual inputs, while only one can actively shape that stream by directing the pair's movements (Fig. 1, top-left). By showing that, despite identical visual experiences, only the actively moving subject acquired adequate visual acuity, the experiment established the importance of active locomotion in learning vision. Analogously, we introduce the 'Tandem RL' setup, pairing an 'active' and a 'passive' agent in a training loop where only the active agent drives data generation, while both perform identical learning updates from the generated data[1]. By decoupling learning dynamics from its impact on data generation, while preserving the non-stationarity of the online learning setting, this experimental paradigm promises to be a valuable analytic tool for the precise empirical study of RL algorithms.

Holding architectures, losses, and crucially data distribution equal across the active and passive agents, or varying them in a controlled manner, we perform a detailed empirical analysis of the failure modes of passive (i.e. non-interactive, offline) learning, and pinpoint the contributing factors in properties of the data distribution, function approximation and learning algorithm. Our study confirms some past intuitions for the failure modes of offline learning, while refining and extending the findings in the deep RL case. In particular, our results indicate an empirically less critical role for bootstrapping than previously hypothesized, while foregrounding erroneous extrapolation or over-generalization by a function approximator trained on an inadequate data distribution as the crucial challenge. Among other things, our experiments draw a sharp boundary between the mostly well-behaved (and analytically well-studied) case of *linear* function approximation, and the *non-linear* case for which theoretical guarantees are lacking. Moreover, we delineate different, more and less effective, ways of enhancing the training data distribution to support successful offline learning, e.g. by analysing the impact of dataset size and diversity, the stochasticity of the data generating policy, or small amounts of self-generated data. Our results provide hints towards a hypothesis relevant in both offline and online RL: robust learning of control with function approximation may require interactivity not merely as a data gathering mechanism, but as a counterbalance to a (sufficiently expressive) approximator's tendency to 'exploit gaps' in an arbitrary fixed data distribution by excessive extrapolation.

---

[1]The 'tandem' analogy is of course that of two riders, both of whom experience the same route, while only the front rider gets to decide on direction.

## 2 The Experimental Paradigm of Tandem Reinforcement Learning

The Tandem RL setting, extending a similar analytic setup in [Fujimoto et al., 2019b], consists of two learning agents, one of which (the 'active agent') performs the usual online training loop of interacting with an environment and learning from the generated data, while the other (the 'passive agent') learns solely from data generated by the active agent, while only interacting with the environment for evaluation. We distinguish two experimental paradigms (see Fig. 1, top-right):

**Tandem:** Active and passive agents start with independently initialized networks, and train on an identical sequence of training batches in the exact same order.

**Forked Tandem:** An agent is trained for a fraction of its total training budget. It is then 'forked' into active and passive agents, which start out with identical network weights. The active agent is 'frozen', i.e. receives no further training, but continues to generate data from its policy. The passive agent is trained on this generated data for the remainder of the training budget.

### 2.1 Implementation

Our basic experimental agent is 'Tandem DQN', an active/passive pair of Double-DQN agents[2] [van Hasselt et al., 2016]. Following the usual training protocol [Mnih et al., 2015], the total training budget is 200 iterations, each of which consists of 1M steps taken on the environment by the active agent, interspersed with regular learning updates (on one, or concurrently on both agents, depending on the paradigm), on batches of transitions sampled from the active agent's replay buffer. Both agents are independently evaluated on the environment for 500K steps after each training iteration.

Most of our experiments are performed on the Atari domain [Bellemare et al., 2013], using the exact algorithm and hyperparameters from [van Hasselt et al., 2016]. We use a fixed set of four representative games to demonstrate most of our empirical results, two of which (BREAKOUT, PONG) can be thought of as easy and largely solved by baseline agents, while the others (SEAQUEST, SPACE INVADERS) have non-trivial learning curves and remain challenging. Unless stated otherwise, all results show averages over at least 5 seeds, with confidence intervals indicating variation over seeds. In comparative plots, **boldface** entries indicate the default Tandem DQN configuration, and gray lines always correspond to the active agent's performance.

### 2.2 The Tandem Effect

We begin by reproducing the striking observation in [Fujimoto et al., 2019b] that the passive learner generally fails to adequately learn from the very data stream that is demonstrably sufficient for its architecturally identical active counterpart; we refer to this phenomenon as the 'tandem effect' (Fig. 1, bottom). We ascertain the generality of this finding by replicating it across a broad suite of environments and agent architectures: Double-DQN on 57 Atari environments (Appendix Figs. 10 & 11), adapted agent variants on four Classic Control domains from the OpenAI Gym library [Brockman et al., 2016] and the MinAtar domain [Young and Tian, 2019] (Appendix Figs. 12 & 15), and the distributed R2D2 agent [Kapturowski et al., 2019] (Appendix Fig. 14). Details on agents and environments are provided in the Appendix[3].

Empirically, we make the informal observation that while active and passive Q-networks tend to produce similar values for typical state-action pairs under the active policy (where the action is the active Q-value function's argmax for a given state), their values are less correlated for other (non-argmax) actions, and in fact the active and passive greedy policies of a Tandem DQN tend to disagree in a large fraction of states under the behavior distribution (on average $> 75\%$ of states, after 100M steps of training, across 57 Atari games; see Appendix Fig. 13). Moreover, in a fraction ($\approx 12/57$) of Atari games, we observe the passive agent's network to strongly over-estimate a fraction of state-action values, with the over-estimation growing as training progresses.

---

[2]Our choice of Double-DQN as a baseline is motivated by its relatively strong performance and robustness compared to vanilla DQN [Mnih et al., 2015], paired with its simplicity compared to later variants like Rainbow [Hessel et al., 2018], which allows for more easily controlled experiments with fewer moving parts.

[3]We provide two Tandem RL implementations: https://github.com/deepmind/deepmind-research/tree/master/tandem_dqn based on the DQN Zoo [Quan and Ostrovski, 2020], and https://github.com/google/dopamine/tree/master/dopamine/labs/tandem_dqn based on the Dopamine library [Castro et al., 2018].

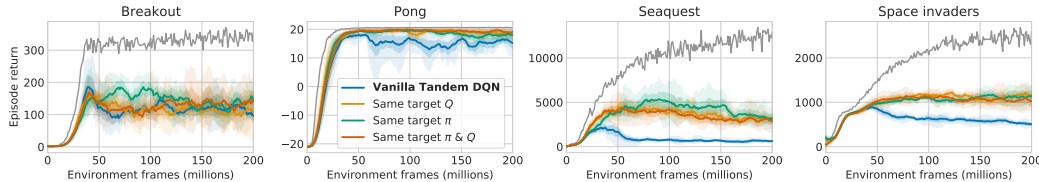

Figure 2: Active vs. passive performance when using the active agent's target policy and/or value function for constructing passive bootstrapping targets.

## 3 Analysis of the Tandem Effect

In line with existing explanations [Levine et al., 2020], we propose that the tandem effect is primarily caused by extrapolation error when certain state-action pairs are under-represented in the active agent's behavior data. Specifically with $\varepsilon$-greedy policies, even small over-estimation of the values of rarely seen actions can lead to sufficient behavior deviations to cause catastrophic under-performance.

We further extend this hypothesis: in the context of *deep* reinforcement learning (i.e. with non-linear function approximation), an inadequate data distribution can drive over-generalization [Bengio et al., 2020], making such erroneous extrapolation likely. While the tandem effect can show up as learning inefficiency even in the tabular case [Xiao et al., 2021], it proves especially pernicious in the case of non-linear function approximation, where erroneous extrapolation can lead to errors not just on completely unseen, but also rarely seen data, and can persist in the infinite-sample limit.

Coalescing this view and past analyses of challenges in offline RL (e.g. [Levine et al., 2020, Fujimoto et al., 2019b, Liu et al., 2020]) into the following three potential contributing factors in the tandem effect provides a natural structure to our analysis:

**Bootstrapping (B)** *The passive agent's bootstrapping from poorly estimated (in particular, over-estimated) values causes any initially small mis-estimation to get amplified.*

**Data Distribution (D)** *Insufficient coverage of sub-optimal actions under the active agent's policy may lead to their mis-estimation by the passive agent. In the case of over-estimation, this may lead to the passive agent's under-performance.*

**Function Approximation (F)** *A non-linear function approximator used as a Q-value function may tend to wrongly extrapolate the values of state-action pairs underrepresented in the active agent's behavior distribution. This tendency can be inherent and persistent, in the sense of being independent of initialization and not being reduced with increased training on the same data distribution.*

These proposed contributing factors are not at all mutually exclusive; they may interact in causing or exacerbating the tandem effect. Insufficient coverage of sub-optimal actions under the active agent's behavior distribution (D) may lead to insufficient constraint on the respective values, which allows for effects of erroneous extrapolation by a function approximator (F). Where this results in over-estimation, the use of bootstrapping (B) carries the potential to 'pollute' even well-covered state-action pairs by propagating over-estimated values (especially via the max operator in the case of Q-learning). In the next sections we empirically study these three factors in isolation, to establish their actual roles and relative contributions to the overall difficulty of passive learning.

### 3.1 The Role of Bootstrapping

One distinguishing feature of reinforcement learning as opposed to supervised learning is its frequent use of learned quantities as preliminary optimization targets, most prominently in what is referred to as *bootstrapping* in the widely used TD algorithms [Sutton, 1988], where preliminary estimates of the value function are used as update targets. In the Double-DQN algorithm these updates take the form $Q(s, a) \leftarrow r + \gamma \bar{Q}(s', \arg\max_{a'} Q(s', a'))$, where $Q$ denotes the parametric Q-value function, and $\bar{Q}$ is the target network Q-value function, i.e. a time-delayed copy of $Q$.

Four value functions are involved in the active and passive updates of Tandem DQN: $Q_A, \bar{Q}_A, Q_P$ and $\bar{Q}_P$, where the $A/P$ subscripts refer to the Q-value functions of the active and passive agents,

respectively. The use of its own target network by the passive agent makes bootstrapping a plausible strong contributor to the tandem effect. To test this, we replace the target values and/or policies in the update equation for the *passive* agent, with the values provided by the *active* agent's value functions:

$$Q_P(s,a) \leftarrow \begin{cases} r + \gamma \bar{Q}_P(s', \arg\max_{a'} Q_P(s', a')) & \text{Vanilla Tandem DQN} \\ r + \gamma \bar{Q}_A(s', \arg\max_{a'} Q_P(s', a')) & \text{Same Target } Q \\ r + \gamma \bar{Q}_P(s', \arg\max_{a'} Q_A(s', a')) & \text{Same Target } \pi \\ r + \gamma \bar{Q}_A(s', \arg\max_{a'} Q_A(s', a')) & \text{Same Target } \pi \& Q \end{cases}$$

As shown in Fig. 2, the use of the active value functions as targets reduces the active-passive gap by only a small amount. Note that when both active target values and policy are used, both networks are receiving *an identical sequence of targets* for their update computations, a sequence that suffices for the active agent to learn a successful policy. Strikingly, despite this the tandem effect appears largely preserved: in all but the easiest games (e.g. PONG[4]) the passive agent fails to learn effectively.

To more precisely understand the effect of bootstrapping with respect to a potential value over-estimation by the passive agent, in Appendix Fig. 16 we also show the values of the passive networks in the above experiment compared to those of the respective active networks. As hypothesised, we observe that the vanilla tandem setting leads to significant value over-estimation, and that indeed bootstrapping plays a substantial role in amplifying the effect: passive networks trained using the active network's bootstrap targets do not over-estimate compared to the active network at all.

These findings indicate that a simple notion of value over-estimation itself is not the fundamental cause of the tandem effect, and that **(B) plays an amplifying, rather than causal role**. Additional evidence for this is provided below, where the tandem effect occurs in a purely supervised setting without bootstrapping.

## 3.2 The Role of the Data Distribution

The critical role of the data distribution for offline learning is well established [Fujimoto et al., 2019b, Jacq et al., 2019, Liu et al., 2020, Wang et al., 2021]. In particular, Wang et al. [2020a] showed that simpler notions of state-space coverage may not suffice for efficient offline learning with function approximation (even in the linear case and under a strong realizability assumption); much stronger assumptions on the data distribution, typically not satisfied in practical scenarios, may actually be required. Here we extend past analysis empirically, by investigating how properties of the data distribution (e.g. stochasticity, stationarity, the size and diversity of the dataset, and its proximity to the passive agent's own behavior distribution) affect its suitability for passive learning.

**The exploration parameter $\varepsilon$**   A simple way to affect the data distribution's state-action coverage (albeit in a blunt and uniform way) is by varying the exploration parameter $\varepsilon$ of the active agent's $\varepsilon$-greedy behavior policy (for training, not for evaluation). Note that a higher $\varepsilon$ parameter affects the *active* agent's own training performance, as its ability to navigate environments requiring precise control is reduced. In Fig. 3 (top) we therefore report the *relative* passive performance (i.e. as a fraction of the active agent's performance, which itself also varies across parameters), with absolute performance plots included in the Appendix for completeness (Fig. 17). We observe that the relative passive performance is indeed substantially improved when the active behavior policy's stochasticity (and as a consequence its coverage of non-argmax actions along trajectories) is increased, and conversely it reduces with a greedier behavior policy, **providing evidence for the role of (D)**.

**Sticky actions**   An alternative potential source of stochasticity is the environment itself, e.g. the use of 'sticky actions' in Atari [Machado et al., 2018]: with fixed probability, an agent action is ignored (and the previous action repeated instead). This type of environment-side stochasticity should not be expected to cause new actions to appear in the behavior data, and indeed Fig. 3 (bottom) shows no substantial impact on the tandem effect.

**Replay size**   Our results contrast with the strong offline RL results in [Agarwal et al., 2020]. We hypothesize that the difference is due to the vastly different dataset size (full training of 200M

---

[4]PONG is an outlier in that it only has 3 actions, and in a large fraction of states actions have no (irreversible) consequences, making greedy policies somewhat robust to errors in the underlying value function.

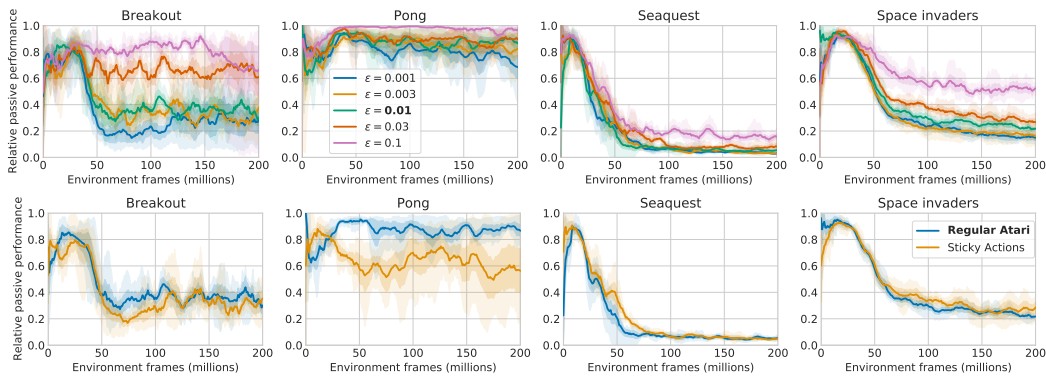

Figure 3: Passive as fraction of active performance for varying active $\varepsilon$-greedy behavior policies (**top**); regular Atari vs sticky-actions Atari (**bottom**). We report *relative* passive performance, as active performance varies across configurations. See Appendix Figs. 17 & 18 for absolute performance.

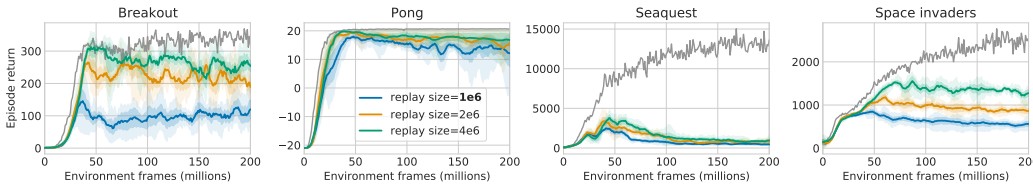

Figure 4: Active vs. passive performance for different replay sizes (for passive agent).

transitions vs. replay buffer of 1M). Interpolating between the tandem and the offline RL setting from [Agarwal et al., 2020], we increase the replay buffer size, thereby giving the passive agent access to somewhat larger data diversity and state-action coverage (this does not affect the active agent's training as the active agent is constrained to only sample from the most recent 1M replay samples, as in the baseline variant). A larger replay buffer somewhat mitigates the passive agent's under-performance (Fig. 4), though it appears to mostly slow down rather than prevent the passive agent from eventually under-performing its active counterpart substantially. As we suspect that a sufficient replay buffer size may depend on the effective state-space size of an environment, we also perform analogous experiments on the (much smaller) classic control domains; results (Appendix Fig. 22) remain qualitatively the same.

Note that for a fixed dataset size, sample diversity can take different forms. Many samples from a single policy may provide better coverage of states on, or near, policy-typical trajectories. Meanwhile, a larger collection of policies, with fewer samples per policy, provides better coverage of many trajectories at the expense of lesser coverage of small deviations from each. To disentangle the impact of these modalities, while also shedding light on the role of stationarity of the distribution, we next switch to the 'Forked Tandem' variation of the experimental paradigm.

**Fixed policy**    Upon forking, the frozen active policy is executed to produce training data for the passive agent, which begins its training initialized with the active network's weights. Note that this constitutes a stronger variant of the tandem experiment. At the time of forking, the agents do not merely share analogous architectures and equal 'data history', but also identical network weights (whereas in the simple tandem setting, the agents were distinguished by independently initialized networks). Moreover, the data used for passive training can be thought of as a 'best-case scenario': generated by a single fixed policy, identical to the passive policy at the beginning of passive training. Strikingly, the tandem effect is not only preserved but even exacerbated in this setting (Fig. 5, top): after forking, passive performance decays rapidly in all but the easiest games, despite starting from a near-optimally performing initialization. This re-frames the tandem effect as not merely the difficulty of passively *learning to act*, but even to passively *maintain performance*. Instability appears to be inherent in the very learning process itself, providing strong support to the hypothesis that an **interplay between (D) and (F) is critical to the tandem effect**.

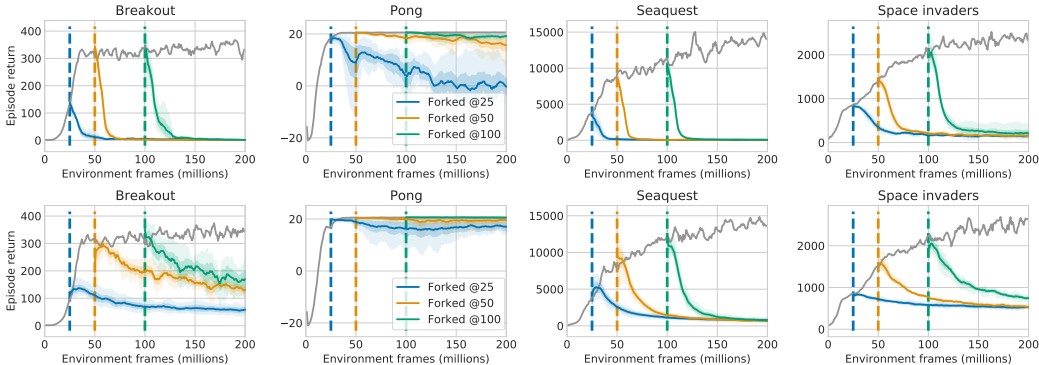

Figure 5: Performance of a Forked Tandem DQN, training passively after freezing its data generating policy (**top**) or its replay buffer (**bottom**). Vertical lines indicate forking time points.

In Appendix Fig. 23 we additionally show that similarly to the regular tandem setting, stochasticity of the active policy *after forking* influences the passive agent's ability to maintain performance.

**Fixed replay**  A variation on the above experiments is to freeze the *replay buffer* while continuing to train the passive policy from this fixed dataset. Instead of a stream of samples from a single policy, this fixed data distribution now contains a fixed number of samples from a training process of the length of the replay buffer, i.e. from a number of different policies. The collapse of passive performance here (Fig. 5, bottom) is less rapid, yet qualitatively similar. In Appendix Fig. 24 we present yet another variant of this experiment with similar results, showing that the effect is robust to minor variations in the exact way of fixing the data distribution of a learning agent.

These experiments provide **strong evidence for the importance of (D)**: a larger replay buffer, containing samples from more diverse policies, can be expected to provide an improved coverage of (currently) non-greedy actions, reducing the tandem effect. While the forked tandem begins passive learning with the seemingly advantageous high-performing initialization, state-action coverage is critically limited in this case. In the frozen-policy case, a large number of samples from the very same $\varepsilon$-greedy policy can be expected to provide very little coverage of non-greedy actions, while in the frozen-replay case, a smaller number of samples from multiple policies can be expected to only do somewhat better in this regard. In both cases the tandem effect is highly pronounced.

**On-policy evaluation**  The strength of the last two experiments lies in the observation that, since active and passive networks have identical parameter values at the beginning of passive training, their divergence cannot be attributed to small initial differences getting amplified by training on an inadequate data distribution. With so many factors held fixed, the collapse of passive performance when trained on the very data distribution produced by *its own initial policy* begs the question whether off-policy Q-learning itself is to blame for this failure mode, e.g. via statistical over-estimation bias introduced by the max operator [van Hasselt, 2010]. Here we provide a negative answer, by performing on-policy evaluation with SARSA [Rummery and Niranjan, 1994] (Fig. 6), and even purely supervised regression on the Monte-Carlo returns (Appendix Fig. 25), in the forked tandem setup. While evaluation succeeds, in the sense of minimizing evaluation error on the given behavior distribution, atypical action values under the behavior policy suffer substantial estimation error, resulting in occasional over-estimation. The resulting $\varepsilon$-greedy control policy under-performs the initial policy at forking time as catastrophically as in the other forked tandem experiments (more details in Appendix A.3). **Strengthening the roles of (D) and (F) while further weakening that of (B)**, these observations point to an inherent instability of offline learning, different from that of Baird's famous example [Baird, 1995] or the 'Deadly Triad' [Sutton and Barto, 2018, van Hasselt et al., 2018]; an instability that results purely from erroneous extrapolation by the function approximator, when the utilized data distribution does not provide adequate coverage of relevant state-action pairs.

**Self-generated data**  Our final empirical question in this section is 'How much data *generated by the passive agent* is needed to correct for the tandem effect?'. While a full investigation of this question exceeds the scope of this paper and is left for future work, the tandem setup lends itself to a

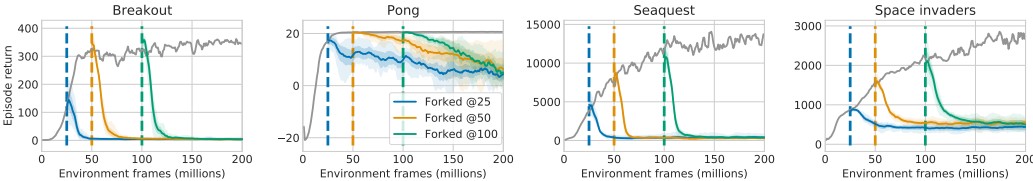

Figure 6: Passive performance in Forked Tandem DQN after policy evaluation with SARSA.

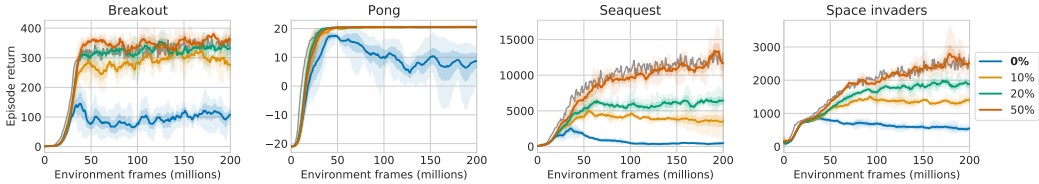

Figure 7: Passive performance for different amounts of self-generated data in the passive agent's replay batches.

simple experiment: *both* agents interact with the environment and fill individual replay buffers, one of them (for simplicity still referred to as 'passive') however learns from data stochastically mixed from both replay buffers. Fig. 7 shows that even a moderate amount (10%-20%) of 'own' data yields a substantial reduction of the tandem effect, while a 50/50 mixture completely eliminates it.

## 3.3 The Role of Function Approximation

We structure our investigation of the role of function approximation in the tandem effect into two categories: the *optimization* process and the *function class* used.

**Optimization**    Agarwal et al. [2020] and Obando-Ceron and Castro [2021] demonstrated that the Adam optimization algorithm [Kingma and Ba, 2015] outperforms RMSProp [Tieleman and Hinton, 2012] used in our experiments. In Appendix Fig. 20 we show that while both active and passive agents perform better with Adam, the tandem effect itself is unaffected by the choice of optimizer.

Another plausible hypothesis is that the passive network suffers from under-fitting and requires more updates on the same data to attain comparable performance to the active learner. Varying the number of passive agent updates per active agent update step, we find that more updates *worsen* the performance of the passive agent (Appendix Fig. 21). This rejects insufficient training as a possible cause, and further **supports the role of (D)**. We also note that, together with the forked tandem experiments in the previous section, this finding distinguishes the tandem effect from the issue of estimation error in the offline learning setting [Xiao et al., 2021]: while in the tabular setting estimation error dominates the learning challenge and a sufficient training duration (assuming full state-space coverage) guarantees convergence to a good solution, this is not necessarily the case with function approximation trained on a given data distribution.

**Function class**    Given that the active and passive agents share an identical network architecture, the passive agent's under-performance cannot be explained by an insufficiently expressive function approximator. Performing the tandem experiment with pure regression of the passive network's outputs towards the active network's (a variant of *network distillation* [Hinton et al., 2015]), instead of TD based training, we observe that the performance gap is indeed vastly reduced and in some games closed entirely (see Appendix Fig. 19); however, strikingly, it remains in some.

Next, we vary the function class of both networks by varying the depth and width of the utilized Q-networks on a set of Classic Control tasks. As can be seen in Fig. 8 (and Appendix Fig. 28), the magnitude of the active-passive performance gap appears negatively correlated with network width, which is **in line with (F):** an increase in network capacity results in less pressure towards over-generalizing to infrequently seen action values and an ultimately smaller tandem effect. On the other hand, the gap seems to correlate *positively* with network depth. We speculate that this may

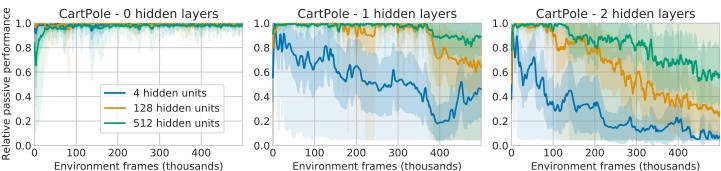

Figure 8: Passive performance as a fraction of active performance in CartPole: varying number of hidden layers and units.

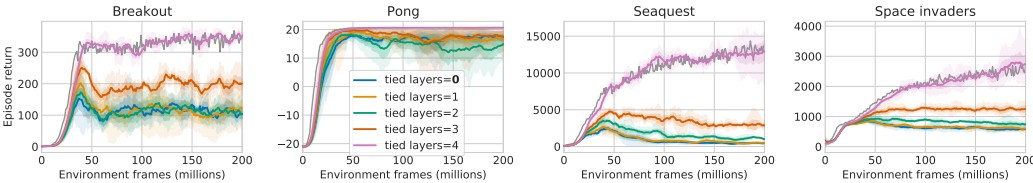

Figure 9: Active vs. passive performance, with first $k$ of 5 layers of active/passive networks shared.

relate to the finding that deeper networks tend to be biased towards simpler (e.g. lower rank) solutions, which may suffer from increased over-generalization [Huh et al., 2021, Kumar et al., 2020a].

Finally, we investigate varying the function class of *only the passive network* by sharing the weights of the first $k$ (out of 5) layers of active and passive networks, while constraining the passive network to only update the remaining top $5 - k$ layers, and using the 'representation' at layer $k$ acquired through active learning only. This reduces the 'degrees of freedom' of the passive agent, which we hypothesize reduces its potential for divergence. Indeed, Fig. 9 illustrates that passive performance correlates strongly with the number of *tied* layers, with the variant for which only the linear output layer is trained passively performing on par with the active agent. A similar result is obtained in the forked tandem setting, see Appendix Fig. 27. This finding provides a strong indirect **hint towards (F)**: with only part of the network's layers being trained passively, much of its 'generalization capacity' is shared between active and passive agents. States that are not aggregated by the shared bottom layers (only affected by active training) have to be 'erroneously' aggregated by the remaining top layers of the network for over-generalization to occur. A more thorough investigation of this, exceeding the scope of this paper, may involve attempting to measure (over-)generalization more directly, e.g. via *gradient interference* [Bengio et al., 2020].

## 4 Applications of the Tandem Setting

In addition to being valuable for studying the challenges in offline RL, we propose that the Tandem RL setting provides analytic capabilities that make it a useful tool in the empirical analysis of general (online) reinforcement learning algorithms. At its core, the tandem setting aims to decouple learning dynamics from its impact on behavior and the data distribution, which are inseparably intertwined in the online setting. While classic offline RL achieves a similar effect, as an analytic tool it has the potential downside of typically using a stationary distribution. Tandem RL, on the other hand, presents the passive agent with a data distribution which realistically represents the type of non-stationarity encountered in an online learning process, while still holding that distribution independent from the learning dynamics being studied. This allows Tandem RL to be used to study, e.g., the impact of variations in the learning algorithm on the quality of a learned representation, without having to control for the indirect confounding effect of a different behavior causing a different data distribution.

While extensive examples of this exceed the scope of this paper, Appendix A.5 contains a single such experiment, testing QR-DQN [Dabney et al., 2018] as a passive learning algorithm (the active agent being a Double-DQN). This is motivated by the observation of Agarwal et al. [2020], that QR-DQN outperforms DQN in the offline setting. QR-DQN indeed appears to be a nontrivially *different* passive learning algorithm, significantly better in some games, while curiously worse in others (Fig. 29).

# 5 Discussion and Conclusion

At a high level, our work can be viewed as investigating the issue of (in)compatibility between the data distribution used to train a function approximator and the data distribution relevant in its evaluation. While in supervised learning, *generalization* can be viewed as the problem of transfer from a training to a (given) test distribution, the fundamental challenge for control in reinforcement learning is that the test distribution is *created* by the very outcome of learning itself, the learned policy. The various difficulties of learning to act from offline data alone throw into focus the role of interactivity in the learning process: only by continuously interacting with the environment does an agent gradually 'unroll' the very data on which its performance will be evaluated.

This need not be an obstacle in the case of *exact* (i.e. tabular) functions: with sufficient data, extrapolation error can be avoided entirely. In the case of function approximation however, as small errors compound rapidly into a difference in the underlying state distribution, significant divergence and, as this and past work demonstrates, ultimately catastrophic under-performance can occur. Function approximation plays a two-fold role here: (1) being an approximation, it allows deviations in the *outputs*; (2) as the learned quantity, it is (especially in the non-linear case) highly sensitive to variations in the *input distribution*. When evaluated for control after offline training, these two roles combine in a way that is 'unexplored' by the training process: minor *output* errors cause a drift in behavior, and thereby a drift in the test distribution.

While related, this challenge is subtly different from the well-known divergence issues of off-policy learning with function approximation, demonstrated by Baird's famous counterexample [Baird, 1995] (see also [Tsitsiklis and Van Roy, 1996]) and conceptualized as the Deadly Triad [Sutton and Barto, 2018, van Hasselt et al., 2018]. While these depend on bootstrapping as a mechanism to cause a feedback-loop resulting in value divergence, our results show that the offline learning challenge persists even without bootstrapping, as small differences in behavior cause a drift in the 'test distribution' itself. Instead of a *training-time* output drift caused by bootstrapping, the central role is taken by a *test-time* drift of the state distribution caused by the interplay of function approximation and a fixed data distribution (as opposed to dynamically self-generated data).

Our empirical work highlights the importance of interactivity and 'learning from your own mistakes' in learning control. Starting out as an investigation of the challenges in offline reinforcement learning, it also provides a particular viewpoint on the classical online reinforcement learning case. Heuristic explanations for highly successful deep RL algorithms like DQN, based on intuitions from (e.g.) approximate policy iteration, need to be viewed with caution in light of the apparent hardness of a policy improvement step based on approximate policy evaluation with a function approximator.

Finally, the forked tandem experiments show that even high-performing initializations are not robust to a collapse of control performance, when trained under their own (*but fixed!*) behavior distribution. Not just *learning to act*, but even *maintaining performance* appears hard in this setting. This provides an intuition that we distill into the following working conjecture: *The dynamics of deep reinforcement learning for control are unstable on (almost) any **fixed** data distribution.*

Expanding on the classical on- vs. off-policy dichotomy, we propose that indefinitely training on *any fixed* data distribution, without strong explicit regularization or additional inductive bias, gives rise to 'exploitation of gaps in the data' by a function approximator, akin to the over-fitting occurring when over-training on a fixed dataset in supervised learning. Interaction, i.e. generating at least moderate amounts of one's own experience, appears to be a powerful, and for the most part *necessary*, regularizer and stabilizer for learning to act, by creating a dynamic equilibrium between optimization of a function approximator and its own data-generation process.

**Broader impact statement**  This work lies in the realm of foundational RL, contributing to the fundamental understanding and development of RL algorithms, and as such is far removed from ethical issues and direct societal consequences. On the other hand, it highlights the empirical difficulty and limitations of offline deep RL for control - increasingly important for practical applications, e.g. robotics, where interactive data is costly, and learning from offline datasets is desirable. In this way it complements existing theoretical hardness results in this area and provides additional context to existing empirical techniques which aim to overcome or circumvent those limitations. We believe that an improved understanding of these challenges can play an important role in creating robust and stable offline learning algorithms whose outputs can be more safely deployed in the real world.

**Acknowledgements**

We would like to thank Hado van Hasselt and Joshua Greaves for feedback on an early draft of this paper, and Zhongwen Xu for an unpublished related piece of work at DeepMind that inspired some of our experiments. We also thank Clare Lyle, David Abel, Diana Borsa, Doina Precup, John Quan, Marc G. Bellemare, Mark Rowland, Michal Valko, Remi Munos, Rishabh Agarwal, Tom Schaul and Yaroslav Ganin, and many other colleagues at DeepMind and Google Brain for the numerous discussions that helped shape this research.

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
