Figure 10: Tandem DQN: active vs. passive performance across Atari 57 (3 seeds per game).

# A   Appendix

## A.1   Implementation, Hyperparameters and Evaluation Details

The implementation of our main agent, Tandem DQN, is based on the Double-DQN [van Hasselt et al., 2016] agent provided in the DQN Zoo open-source agent collection [Quan and Ostrovski, 2020]. The code uses JAX [Bradbury et al., 2018], and the Rlax, Haiku and Optax libraries [Budden et al., 2020, Hennigan et al., 2020, Hessel et al., 2020] for RL losses, neural networks and optimization

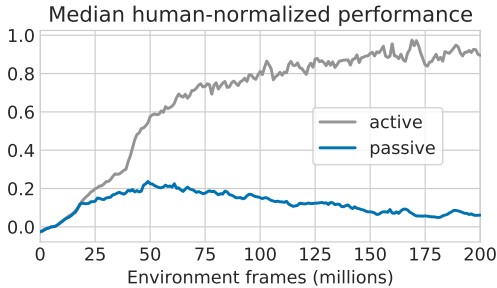

Figure 11: Tandem DQN: Median human-normalized scores over 57 Atari games (3 seeds per game).

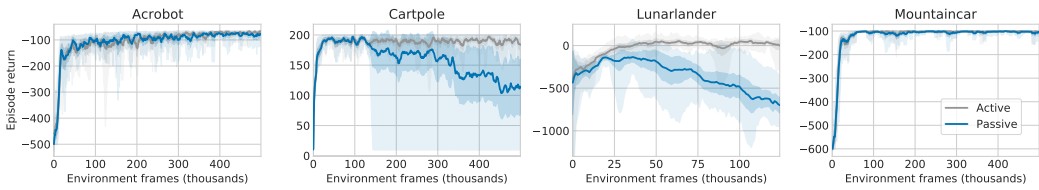

Figure 12: Tandem DQN: Active vs. passive performance on four selected Classic Control domains.

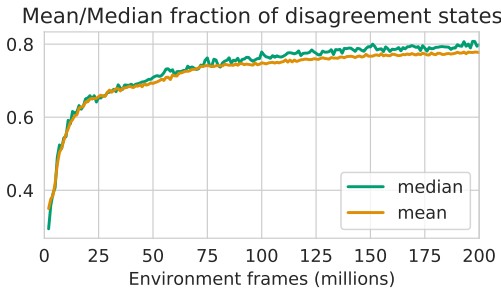

Figure 13: Fraction of states (uniformly sampled from replay buffer) on which active and passive policies disagree, i.e. where $\arg\max_a Q_A(s, a) \neq \arg\max_a Q_P(s, a)$, mean and median across 57 Atari games (3 seeds per game).

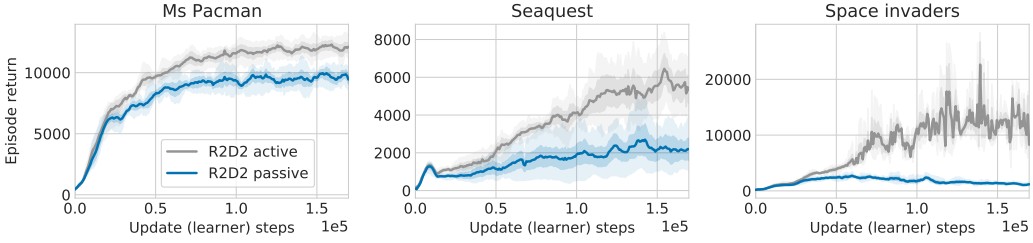

Figure 14: Tandem R2D2: active vs. passive performance on three Atari domains (3 seeds per game). Note: because of the use of an untuned implementation of R2D2, active results are not directly comparable to those of the published agent [Kapturowski et al., 2019].

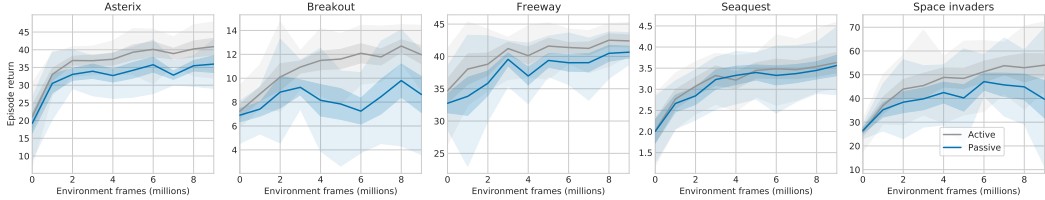

Figure 15: Tandem DQN evaluated on five MinAtar domains.

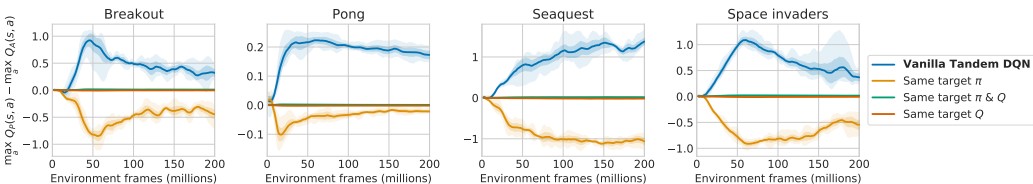

Figure 16: Q-value over-estimation by the passive network compared to the active one in Tandem DQN with varying bootstrap targets. It can be seen that the passive network in the vanilla Tandem DQN setting tends to over-estimate values (compared to the active one), which is almost perfectly mitigated by using the same bootstrap target *values* as the active network, and in fact reversed when using the same target *policy* (but not the same target values). Note that in all four configurations the passive agents substantially under-perform their active counterparts (Fig. 2), showing that bootstrapping-amplified over-estimation is only part, not the main cause of the tandem effect.

algorithms, respectively. All algorithmic hyperparameters correspond to those in the DQN Zoo implementation of Double-DQN.

In the 'Tandem' setting, active and passive agents' networks weights are initialized independently. Agents in this setting are trained in lockstep, i.e. active and passive agents are updated simultaneously from the same batch of sampled replay transitions, with the exception of one experiment in Section 3.3, where we study the effect of the number of passive updates relative to active agent updates.

In the 'Forked Tandem' setting, only one of the agents is trained at any one time. The active agent trains (as a regular Double-DQN) up to the time of forking, at which point the passive agent is created as a 'fork' (i.e., with identical network weights) of the active agent. After forking, only the passive agent is trained. The active agent is used for data generation, either by executing its policy and continuing to fill the replay buffer ('Fixed Policy' experiment), or by sampling batches from its 'frozen' last replay buffer obtained in the active phase of training ('Fixed Replay' experiment).

The total training budget is kept fixed at 200 iterations in both settings, split across 'active' and 'passive' training phases in the forked tandem setting. In all cases, both active and passive agents are evaluated after each training iteration for 500K environment steps. Executed with an NVidia P100 GPU accelerator, each Atari training run takes approximately 4.5 days of wall-clock time.

The majority of our Atari experiments use the regular ALE Atari environment [Bellemare et al., 2013], using DQN's default preprocessing, random noop-starts and action-repeats [Mnih et al., 2015], as well as using the reduced action-set (i.e. each game exposing the subset of Atari's total 18 actions which are relevant to this game). For the 'Sticky actions' experiment, we use the OpenAI Gym variant of Atari [Brockman et al., 2016] enhanced with sticky actions [Machado et al., 2018].

Unless stated explicitly, all our results are reported as mean episode returns averaged across 5 seeds, with light and dark shading indicating $0.5$ standard deviation confidence bounds and min/max bounds (across seeds), respectively. Gray curves always indicate active performance.

The 'relative passive performance' (or 'passive performance as fraction of active performance') curves are meant to illustrate the relative (under-)performance of the passive agent compared to its active counterpart in cases where the active agent's performance varies strongly across configurations. Denoting $R_a(t)$, $R_p(t)$ the active and passive (undiscounted) episodic returns at iteration $t \in$

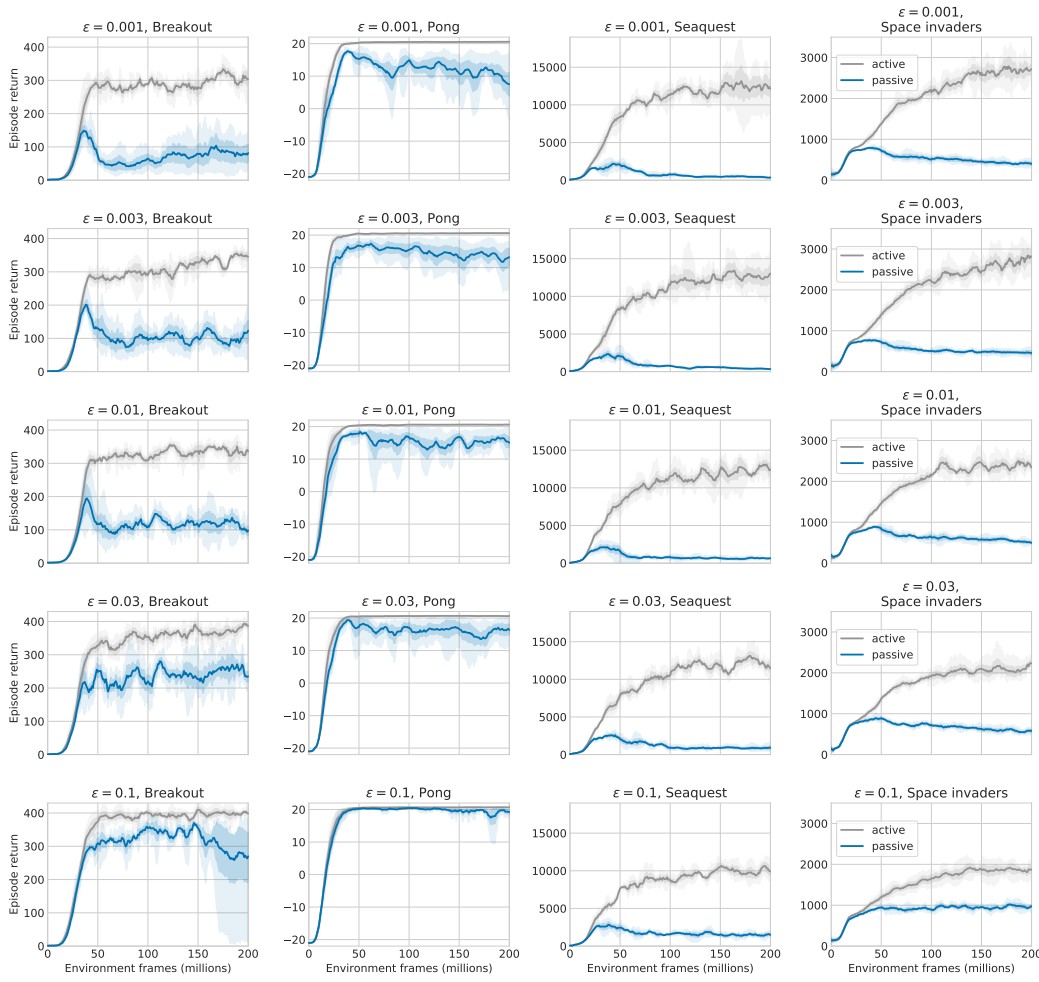

Figure 17: Active vs. passive performance for varying active $\varepsilon$-greedy behavior policies. Note that here *active* performance varies across settings.

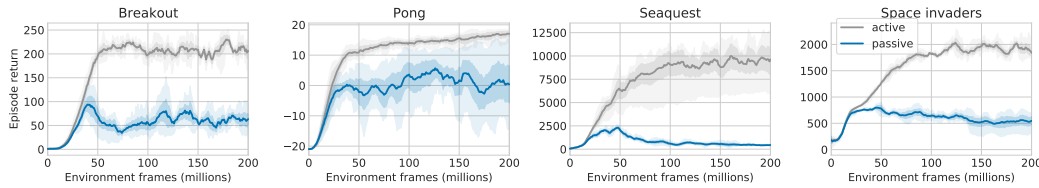

Figure 18: Active vs. passive performance on Atari with sticky actions [Machado et al., 2018].

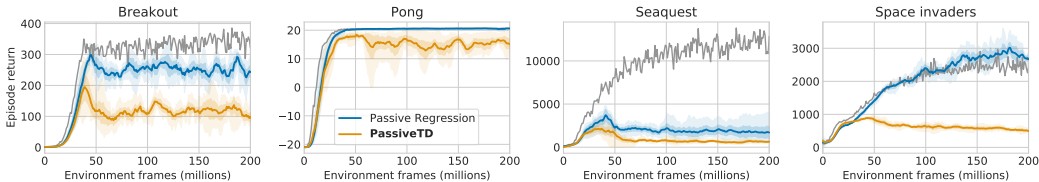

Figure 19: Active vs. passive performance with regular (TD-based) and regression-based Tandem DQN. The latter regresses all the passive agent's action-values towards the respective outputs of the active agent's network, which can be viewed as network distillation.

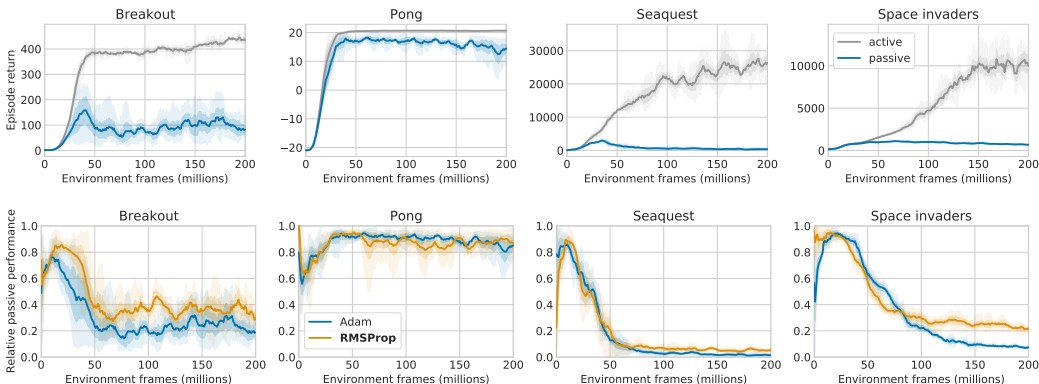

Figure 20: Tandem DQN with the Adam optimizer (instead of RMSProp) for both active and passive network optimization. (**top**) Adam: Active vs. passive performance. (**bottom**) Passive as fraction of active performance Adam vs. RMSProp. While the Adam optimizer improves both the active and passive performance of the Tandem DQN, the relative active-passive gap is not affected strongly.

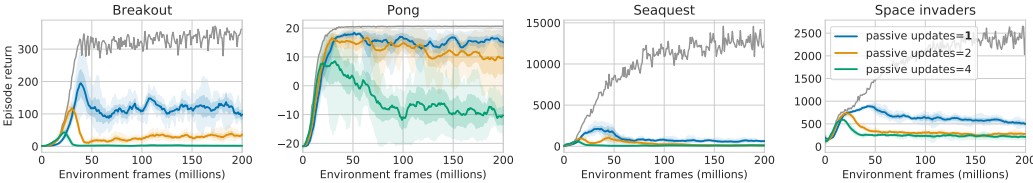

Figure 21: Active vs. passive performance for varying number of passive updates per active update.

$\{0, \ldots, 200\}$, and setting $m = \min_t \min(R_a(t), R_p(t))$, the relative performance is computed as

$$\frac{R_p(t) - m}{R_a(t) - m}$$

with the value being clipped to lie in $[0, 1]$ and set to $1.0$ whenever $R_a(t) = m$.

For the classic control [Brockman et al., 2016] and MinAtar [Young and Tian, 2019] experiments we used a modified version of the DQN agent from the Dopamine library [Castro et al., 2018]. The modifications made were:

- Double-DQN [van Hasselt et al., 2016] learning updates instead of vanilla DQN
- MSE loss instead of Huber loss (as suggested in [Obando-Ceron and Castro, 2021])
- Networks and wrappers for running MinAtar with the Dopamine agents
- Tandem training regime (regular and/or forked) instead of regular single-agent training.

Unless stated explicitly, all hyperparameters follow the respective default configurations in the Dopamine library. Our network architecture for the classic control environments are two fully

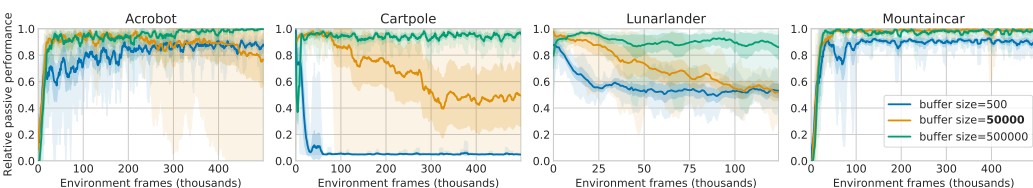

Figure 22: Passive performance as a fraction of active performance when varying the size of the replay buffer used by the passive agent on Classic Control domains.

connected layers of 512 units (each with ReLu activations), followed by a final fully connected layer that yields the Q-values for each action. In Figs. 8 and 28 we varied the number of hidden layers and units, where the variation of number of units is applied uniformly across all layers.

The default network architecture for the MinAtar environments is one convolutional layer with 16 features of $3 \times 3 \times 3$ and stride of 1 followed by a ReLu activation, whose output is mapped via a fully connected layer to the network's Q-value outputs.

The classic control environments were all run on CPUs; each run took between 20 minutes (CARTPOLE) and 2 hours (MOUNTAINCAR). The MinAtar environments were all run on NVidia P100 GPUs, each run taking approximately 12 hours to complete. Results for all classic control and MinAtar environments are reported as mean episode returns averaged across 10 seeds, with light and dark shading indicating $0.5$ standard deviation confidence bounds and min/max bounds (across seeds), respectively.

For the R2D2 experiment (Fig. 14), an untuned variant of the distributed R2D2 algorithm [Kapturowski et al., 2019] was used. Each run used 4 TPUv3 chips for learning and inference, together with a fleet of approximately 500 CPU-based actor threads for distributed environment interaction, completing a training run of approximately 150K batch updates in about 7 hours wall-clock time.

## A.2  Forked Tandem: Variants

Here we present additional experimental variants performed within the Forked Tandem setup.

**Varying exploration parameter $\varepsilon$ with fixed policy:**  This experiment is an extension of the 'Fixed Policy' (Fig. 5) and 'The exploration parameter $\varepsilon$' (Fig. 3 (top)) experiments. After freezing the active agent's policy for further data generation, its $\varepsilon$ parameter is set to a different value, to explore the impact of the resulting policy stochasticity on the ability of the passive learning process to maintain the initial performance level. We note that because of the fixed active policy, in this case active training performance does not depend on the chosen configuration, and so absolute passive performance curves are more directly comparable.

Similar to the results in the regular tandem setup, we observe (in Fig. 17) that the ability of the passive agent to maintain the initial performance level is substantially aided by the stochasticity resulting from a higher value of $\varepsilon$, providing further **support for the importance of (D)**.

**Training process samples ('Groundhog day'):**  The forked tandem experiments in Section 3.2 indicate that data distributions represented by a fixed replay buffer or a stream of data generated by a single fixed policy both show a lack of diversity leading to a catastrophic collapse of an (initially high-performing) agent when trained passively. The naive expectation that the (unbounded) stream of data generated by a fixed policy may provide a better state-action coverage than the fixed-size dataset of a single replay buffer (1M transitions) is invalidated by the observation of the fixed-replay training leading to somewhat slower degradation of passive performance. Unsurprisingly in hindsight, the diversity given by the samples stemming from many different policies along a learning trajectory of 1M training steps appears to be significantly higher than that generated by a single $\varepsilon$-greedy policy.

To probe this further, we devise an experiment attempting to combine both: instead of freezing the active policy after forking, we continue training it (and filling the replay buffer), however after each iteration of 1M steps, the active network is reset to its parameter values at forking time. Effectively this produces a data stream that can be viewed as producing *samples of the training process* of a single iteration, a variant that we refer to as the 'Groundhog day' experiment. This setting combines the multiplicity of data-generating policies with the property of an unbounded dataset being presented to the passive agent. The results are shown in Fig. 24 - indeed we observe that the groundhog day setting improves passive performance over the fixed-policy setting, while not clearly changing the outcome in comparison to the fixed-replay setting.

Overall we observe a general robustness of the tandem effect with respect to minor experimental variations in the forked tandem settings.

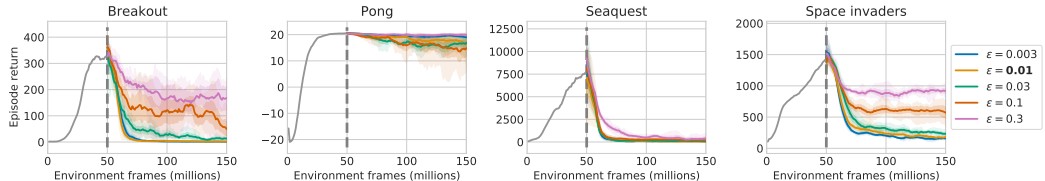

Figure 23: Forked Tandem DQN: After 50 iterations of regular active training, the active value function is frozen and used to continuously generate data for the passive agents' training by executing an $\varepsilon$-greedy policy with a given value of $\varepsilon$.

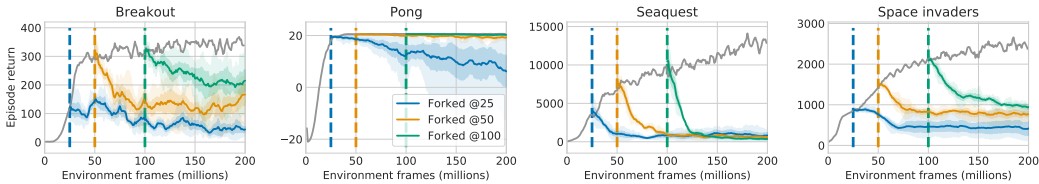

Figure 24: Forked Tandem DQN: 'groundhog day' variation, active vs. passive performance. After forking, the active agent trains for a full iteration (1M environment steps), and is then reset to its initial network parameters at the time of forking, repeatedly for the remaining number of iterations.

### A.3    Forked Tandem: Policy Evaluation

Among the most striking findings in our work are the forked tandem findings in Sections 3.2 and A.2, demonstrating a catastrophic collapse of performance when passively training from a fixed data distribution, even when the starting point for the passive policy is the very same high-performing policy generating the data. This leads to the question whether the process of Q-learning itself is to blame for this failure mode, e.g. via the well-known statistical over-estimation bias introduced by the max operator [van Hasselt, 2010]. To test this, we perform two variants of the forked tandem experiment with SARSA [Rummery and Niranjan, 1994] and (purely supervised) Monte-Carlo return regression based policy evaluation instead of Q-learning as the passive learning algorithm. (We note that while SARSA evaluation of an $\varepsilon$-greedy policy can still exhibit over-estimation bias, this is not the case for Monte-Carlo return regression.)

As can be seen in Figs. 6 and 25(top), even in this on-policy policy evaluation setting, the resulting control performance catastrophically collapses after a short length of training. We also observe that

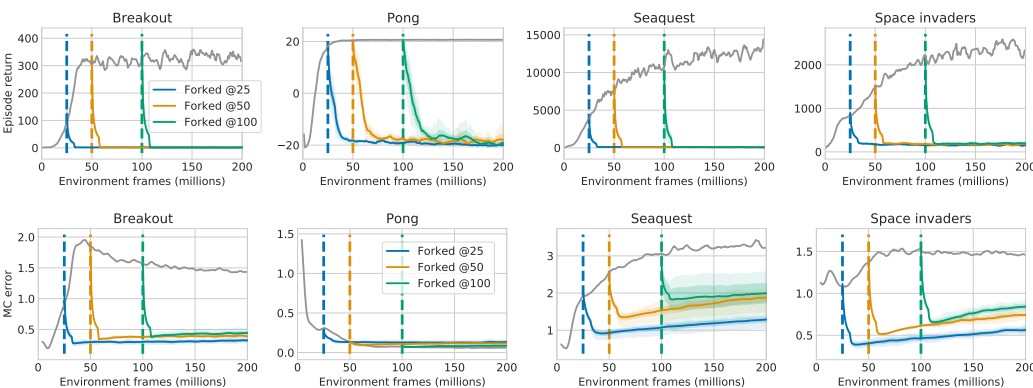

Figure 25: Forked Tandem DQN: Policy evaluation with Monte-Carlo return regression. (**top**) Active vs. passive control performance. (**bottom**) Average absolute Monte-Carlo error. The Monte-Carlo error is minimized effectively by the passive training, while the control performance of the resulting $\varepsilon$-greedy policy collapses completely.

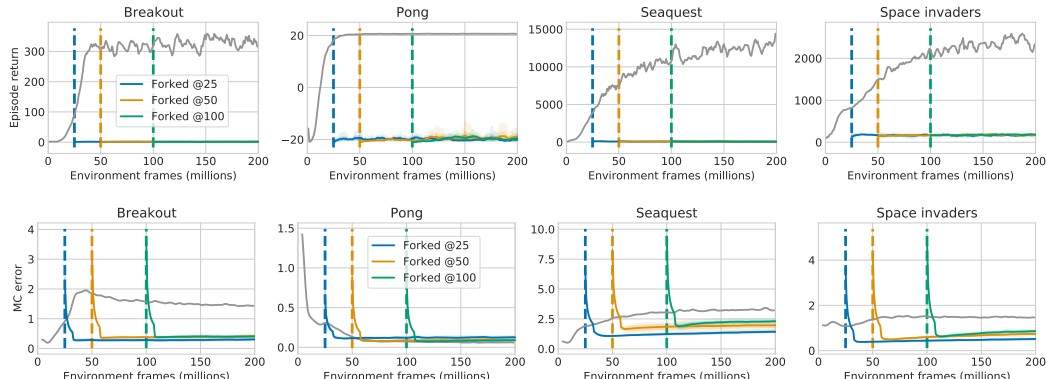

Figure 26: Forked Tandem DQN: Policy evaluation with Monte-Carlo return regression, passive network initialized independently of the active network at forking time. Top: Active vs. passive control performance. Bottom: Average absolute Monte-Carlo error. While Monte-Carlo error is minimized effectively by the passive training, this does not result in above-random control performance of the resulting $\varepsilon$-greedy policy.

this is not due to a failing of the evaluation: Fig. 25(bottom) shows effective minimization of the Monte-Carlo error, indicating that the control failure is due to extrapolation error (and in particular, over-estimation) on infrequent (under the active policy) state-action pairs.

These findings provide strong **support for the role of (D) and (F)** while **weakening that of (B)**: In contrast to the well-known 'Deadly Triad' phenomenon [Sutton and Barto, 2018, van Hasselt et al., 2018], the tandem effect occurs without the amplifying mechanism of bootstrapping or the statistical over-estimation caused by the max operator, solely due to erroneous extrapolation by the function approximator to state-action pairs which are under-represented in the given training data distribution.

We have so far presented the equal network weights of active and passive networks at forking time as a strength of the forked tandem setting, following the intuition that an initialization by the high-performing policy should be advantageous for maintaining performance in these experiments. A plausible counter-argument could be that the representation (i.e., the network weights) learned by the active agent *in service of control* could be a poor, over-specialized starting point for policy evaluation. To verify that this is not a major factor, we also perform the above Monte-Carlo evaluation experiment with the passive network freshly re-initialized at the beginning of passive training. As shown in Fig. 26, while a fresh initialization of the passive network indeed allows it to similarly effectively minimize Monte-Carlo error, its control performance here never exceeds random performance levels, further **connecting the tandem effect to (D) and (F).**

We remark that the demonstrated control performance failure of approximate policy evaluation casts a shadow over the concept of approximate policy iteration, or the application of this concept in heuristic explanations of the function of empirically successful algorithms like DQN [Mnih et al., 2015]. A successful greedy improvement step on an *approximately evaluated* policy appears implausible given the brittleness of approximate policy evaluation *even in the nominally best-case scenario of an on-policy data distribution*.

Another view point emerging from these results is that the classic category of 'on-policy data' appears less relevant in this context: an appropriate data distribution for robust approximate evaluation targeting *control* seems to require a data distribution sufficiently overlapping with the (hypothetical, in practice unavailable) behavior distribution of the *resulting policy* rather than the original evaluated policy.

### A.4 Additional Experimental Results on the Role of Function Approximation

Here we present several extra experiments, complementing the results from the end of Section 3.3.

The first set of results, shown in Fig. 27, concerns the passive performance in the forked tandem setup, when the first (bottom) neural network layers are shared between active and passive agents.

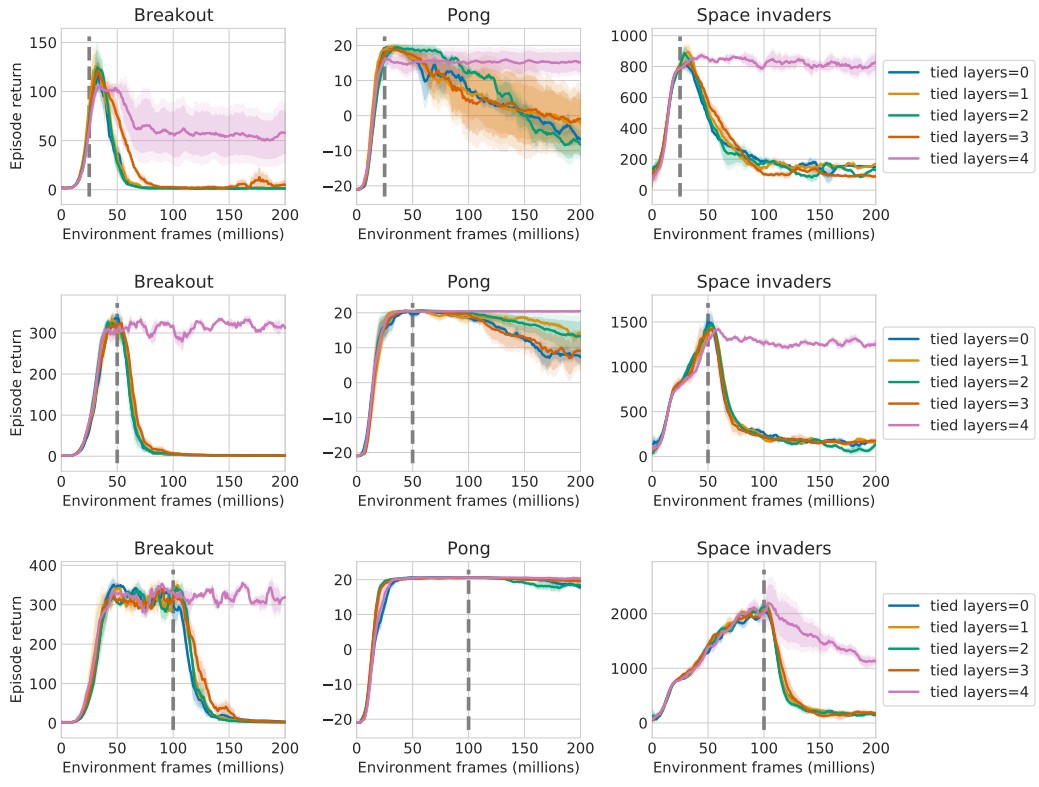

Figure 27: Forked Tandem DQN: passive performance after forking, with the first $k$ (out of 5) layers of active and passive agent networks shared, i.e. only trained by the active agent, and fixed during the passive training phase.

These bottom layers are only trained during the active training phase before forking, and are frozen after that. Similar to the corresponding experiment in the regular tandem setting (Fig. 9), we observe that the passive agent's ability to maintain its initially high performance strongly correlates with the number of *shared*, i.e. not passively trained layers. The difference between the passive agent training a 'deep' vs. a 'linear' network (the latter corresponds to all but one of the network layers being frozen) appears stark: the tandem effect is almost equally catastrophic in all configurations except for the linear one, where it appears to be strongly reduced. While a more thorough investigation remains to future work, we remark that overall this finding appears to **supports the importance of (F)**, in that intuitively over-extrapolation can be expected to become more problematic when passive training is applied to a larger function class (deeper part of the network).

The next experiment, shown in Fig. 28 and extending the results from Fig. 8, investigates the impact of network architecture more generally, by varying width and depth of both active and passive agents' networks. Since changes in Atari-DQN network architecture tend to require expensive re-tuning of various hyperparameters, we chose to perform these experiments on the smaller Classic Control domains, where such changes tend to be more straightforward. Nevertheless active performance in these domains does depend on the chosen network configuration, so that we report relative performance as the more informative quantity. The findings across four domains mostly appear to echo those on CARTPOLE described in Section 3.3, showing a positive correlation of (relative) passive performance with network *depth*, and a negative correlation with its *width*. Again, a more detailed investigation of the causes for this exceeds the scope of this paper and is left to future work.

### A.5 Applications of Tandem RL: Passive QR-DQN

Here we provide an example application for the Tandem RL setting as an analytic tool for the study of learning algorithms in isolation from the confounding factor of behavior. As observed in [Agarwal

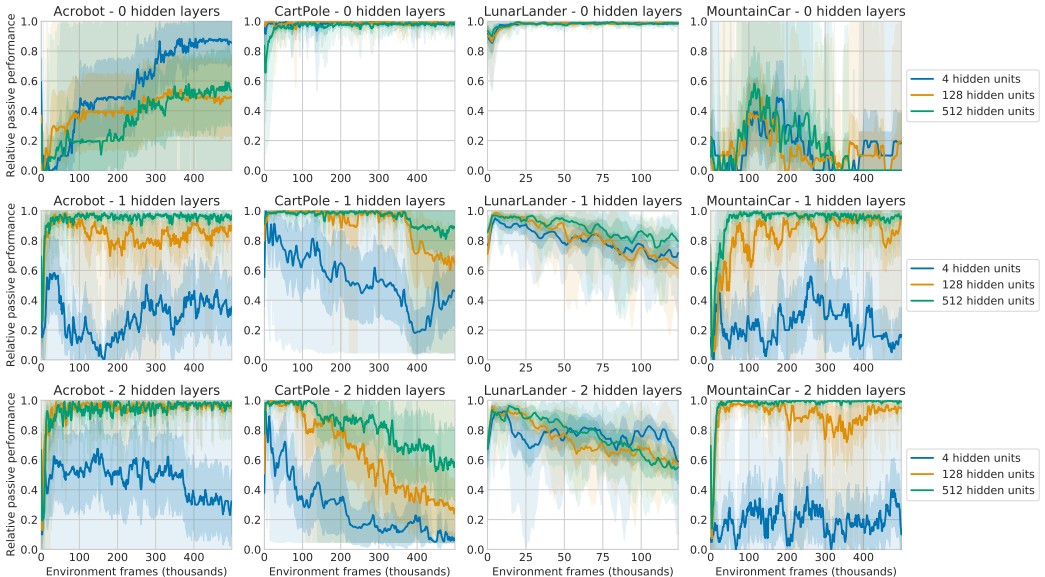

Figure 28: Tandem DQN: Passive performance as a fraction of active performance when varying network architecture in classic control games. Here network architecture varies for both active and passive agent, so active performance is also affected by the configuration.

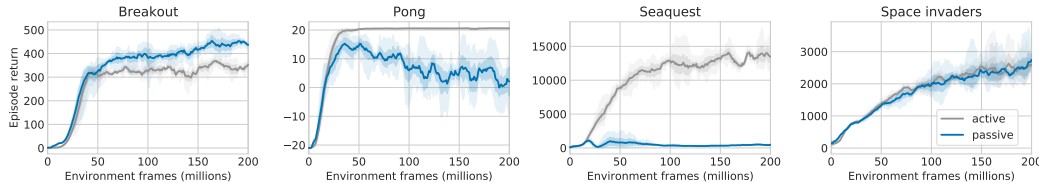

Figure 29: QR-DQN as a passive learning algorithm, in tandem with a Double-DQN active agent: active vs. passive performance.

et al., 2020], the QR-DQN algorithm [Dabney et al., 2018] can be preferable to DQN in the offline setting, motivating our attempt to use it as a passive agent, coupled with a regular Double-DQN active agent. As shown in Fig. 29, QR-DQN indeed provides a somewhat different passive performance profile when compared to the regular Double-DQN tandem, albeit not a clearly better one. While perfectly matching active performance in one game (SPACE INVADERS) and even *out-performing the active agent* in another (BREAKOUT), it also shows exacerbated under-performance or instability in the other two domains. A fine-grained diagnosis of the causes of this are left to future work.

We note that any difference in performance between the DQN and QR-DQN algorithms as passive agents reflects directly on their properties as learning algorithms, i.e. their respective abilities to extract information about an appropriate control policy from observational data, while separating out any influence their learning dynamics may have on (transient) behavior and data generation. We believe that Tandem RL can become a valuable analytic tool for targeted empirical studies of such properties of learning algorithms.