# OpenReview forum: "The Difficulty of Passive Learning in Deep Reinforcement Learning"
_NeurIPS.cc/2021/Conference — NeurIPS 2021 Poster_

### Official Review · Reviewer_LigT · 2021-06-25

**Rating:** 5
**Confidence:** 4

**Summary:**

This paper presents an empirical study on offline deep reinforcement learning in atari environments. The authors use tandem learning -- where an agent B learns from experience collected from another agent A. The authors consider 3 hypotheses as to why agent B tends to perform worse in this setting, and provide many ablation-type experiments to support/reject these hypotheses.

**Main Review:**

Main comments
---------------

Overall the paper is well written and relatively easy to follow. Offline RL is an important domain, and the experimental setup enables the authors to probe many important questions. The authors specify the hypotheses they are testing and interpret what the experiments say about these. This is a good step towards good empirical practices.

Compared to previous work, there are some novel and interesting experiments: 1) the authors consider using different bootstrap targets for the tandem learning, and shows that the tandem-gap persists even if using the bootstrap targets of "online" agent. 2) The authors study the effects of \epsilon and replay buffer size on the tandem gap. The question here is interesting, but the results vary among environments which makes it hard to draw conclusions. 3) A method of "forking" a tandem learning, which demonstrates that the tandem gap appears quickly in this case.

There are two main issues with the paper 1) the novelty of the setup is not clear and 2) the clarity of the hypotheses and what the experiments say could be improved.


**Novelty**

The experimental setup of "tandem learning" seems to already have been proposed by [13](Scott Fujimoto, David Meger, and Doina Precup. Off-policy deep reinforcement learning without exploration.). Indeed, the authors state that "We begin by reproducing the striking observation in [13]". However, in the abstract, the authors state that "we propose the “tandem learning” experimental paradigm". Could the authors clarify their contribution of proposing tandem learning vis-a-vis [13]?


**Clarity**

It would be preferable to state the 3 hypotheses of Section 3 in a mathematical language to make them more precise. It is unclear what is meant by e.g. "the stationarity of the data distribution [...] play only a small role.". What is a "small" role? Since the hypotheses are stated in a relatively "hand-wavy" way, it becomes hard to see what the experiments say about them. Since the hypotheses drive the experiments, it should be crystal clear what they are and how they differ.


In general, only graphs are presented and used to draw conclusions. When making precise statements, it would be preferable to use some specific statistics and then perform statistical significance testing to make sure the results are not random noise. E.g. the authors state that "As can be seen in Fig. 3 (top), relative passive performance is indeed substantially improved".  Fig 3 shows improvements for some games, but differences for seaquest or pong might as well be due to chance. The author states "Unless stated otherwise, all results show averages over at least 5 seeds, with confidence intervals indicating variation over seeds." This appears to be the ONLY place in the text where seeds are mentions. What experiments use more than 5 seeds? And if there are any, how many seeds are used? At any rate, 5 seeds are very low for making statistically significant observations.




Minor Comments
-----------

I doubt most reviewers are familiar with "Held & Hein’s classic 1963 experiment" as stated in the abstract. Referring to it so early will likely confuse most readers.

Many figures lack labeled axes, see e.g. Fig 3,4, and 5.

Is CUDA set to deterministic mode? How are the seeds selected?

Double-DQN, the used architecture, is a relatively simple model that isn't used in practice. I think it would have been better to use rainbow, and simply use hyperparameters from reference implementations. By not changing such parameters, the authors can keep the "moving parts" to a minimum.

Some experiments seem to use only 4 environments. Whenever results for more environments are available, I think they should be presented in the main paper.

## post rebuttal comment

I have read the rebuttal and thank the authors for their response.

My two main objections to the paper remains. Firstly, I don't think the novelty is particularly high. The authors state that a novel aspect is "explicit framing of the Tandem as an analytic tool". The tool has clearly already been used implicitly as an analytic tool in a previous paper. Framing a known method as an explicit analytic tool doesn't count as novel to me. Secondly, I think the clarity is lacking due to the lack of statistical rigor.  The authors state that "given the inner complexity of any modern deep RL algorithm, we have little hope of being able to pin these down with full mathematical rigour". Full rigor might be unattainable, nonetheless I think the paper could be improved significantly along this direction. I believe that the paper would be better served by using more seeds on fewer environments instead of using a few seeds across the whole 57 task Atari suite. However, the paper also has many strong points and to me this is a borderline paper. I am clearly in the minority regarding these opinions, and given the other scores I expect to see the paper in the final program and I'm sure it would bring about many interesting discussions despite the issues given above.







**Time Spent Reviewing:**

1

---

> ### Author Response · Authors · 2021-08-10
> **Response to Reviewer LigT**
>
> Thank you for your review and feedback, your positive assessment of the writing, experimental practice and relevance of our work, and for your comments and questions. We will incorporate your useful suggestions in a revised version of the paper, and attempt to address all of your main questions & comments below:
>
> **Regarding novelty**: it is indeed the case that the 'tandem RL' setting (or perhaps more generally a 'tandem learning' setup) is not completely new, and was briefly used in a similar context by Fujimoto et al [13], and before that in the Held & Hein 1963 work as well as other related work in psychology. We have attempted to be clear & forthcoming about this in the paper, and will make this even clearer in a revised version. What we believe to be a novel aspect is the explicit framing of the Tandem as an analytic tool for the study of learning dynamics in separation from its effects on behaviour & exploration. While on the one hand (as presented in the paper) this allows to obtain sharper empirical results on the difficulties of offline RL and disentangle certain confounding effects, on the other hand it also reaches beyond offline RL, as we tried to highlight in our Section 4 and continued in Appendix A.5. To the best of our knowledge, this experimental protocol (and specifically its two instantiations as the Regular and Forked Tandem) has not been realized as an interesting analytic tool in its own right in the ML field prior to this work.
>
> **Regarding clarity**: we are grateful for the observation, and will attempt to state the hypotheses more unambiguously and crisply in a revised version. On the other hand, given the inner complexity of any modern deep RL algorithm, we have little hope of being able to pin these down with full mathematical rigour. Ultimately our analysis remains empirical (and is intended as such), and the drawn conclusions are mostly of a qualitative nature - that certain factors appear to be causally linked, with a precise quantification (in the case of deep RL on large environments) currently out of reach due to a very large number of confounding interactions in these complex algorithms.
>
> **Regarding the presentation of (mostly) graphs, statistical significance, and number of seeds**: We have used 5 seeds in all of our individual Atari experiments, but only 3 seeds for the experiment involving all 57 Atari games (Fig 9 and 10 in the Appendix) as well as the Tandem  R2D2 results (Fig 13) due to the excessive computational expense - these deviations from the default are noted in the respective Figure captions (in the appendix). Statistical significance claims are generally quite rarely possible in deep RL on large environments, and even 5 seeds across this number of experiments is, as far as we are aware, unusually high for this literature.
>
> **Regarding CUDA**: We do not use deterministic mode, and GPU/CUDA non-determinism is one source of stochasticity in our experiments. The different seeds in our experiments are consecutive integers (1, 2, 3, 4, 5) being passed to PRNG initialization functions of numpy and JAX used in our code. In all the 'regular tandem' experiments, network parameters are initialized independently as two consecutive samples from identical distributions, whereas in the forked-tandem experiments the passive net is not initialized independently but as a direct copy of the active network at forking time.
>
> **Regarding Double-DQN**: While we agree that post-2017 Rainbow has become the dominant algorithm, we would argue that Double-DQN (with its published default hyperparameters) remains among the 'simplest while robust & effective' baselines. In an empirical-analytic work like this, we were striving for maximal simplicity of the underlying algorithm to minimize unconsidered complex algorithmic interactions conflating the findings, while trying to remain in an empirically relevant setting. Any of the components ("prioritized replay", "noisy net exploration", "c51 distributional loss") would introduce more 'moving parts' that may interact in unexpected ways with our ablations and algorithmic variations. Consider for example prioritized replay, which introduces a dependency of the sampling distribution (crucial in our experiments) on the actual value function - in the passive learner this would raise a number of different implementational possibilities with varying trade-offs and a much larger number of necessary ablations. In light of this we believe our choice of baseline remains sensible, though the question whether other learning algorithms are more effective choices in the offline RL setting is an interesting and open one (and can actually be studied well in the tandem setting itself - see Section 4 & Appendix A.5)
>
> **Regarding the number of environments**: Due to the high number of experiments & corresponding computational cost, we have chosen to limit ourselves to 4 environments in most of the experiments, and only included a full Atari-57 run (as well as some alternative agent implementations like R2D2) in the basic Tandem experiment as demonstration of generality of the effect. We included these extra results in the appendix rather than the main text due to space limitations, but mentioned them in the main text.

---

### Official Review · Reviewer_cED7 · 2021-07-04

**Rating:** 8
**Confidence:** 4

**Summary:**

The paper addresses the difficulties that standard RL methods have in the offline setting. It presents a new analysis technique designed to better understand the causes of the difficulties. The technique, called Tandem RL setting, and forked tandem, extend the setup, introduced by Fujimoto, Meger, and Precup in the BCQ-paper "Off-Policy Deep Reinforcement Learning without Exploration" and enable systematic studies of causes for some kind of problems with offline RL. The technique is used for extensive studies, in several benchmarks, and cautious conclusions are drawn.

**Limitations And Societal Impact:**

Yes.

**Main Review:**

Strengths:
It seems to me a valuable direction to strengthen the analysis techniques for understanding the root cause of the problems. This can be used to support the development of methods.

Weaknesses:
None.
One would like to see more. In particular, studies on how the individual techniques developed to make offline RL more robust (batch regularization, batch constraining, use of models, use of uncertainty) perform in this analysis.
But, of course, this is the task for future work, as it is way beyond the scope of this paper.

Originality:
The conception of the paper, as well as the presented method are original.

Quality:
Very good.

Clarity:
Very clear.

Significance:
Very high. I can well imagine that the presented approach to analysis will be used in the future and help to further increase the performance of offline RL methods.

Further comments:
In the paragraph at the end of the 1st page, starting with "To mitigate this," I think the mention of uncertainty consideration as an approach to make offline RL more robust is missing. This technique is used frequently in model-based offline RL, e.g. [1,2,3], but also in model-free offline RL [4].
[1] Decomposition of uncertainty in Bayesian deep learning for efficient and risk-sensitive learning, ICML 2018
[2] MOPO: Model-based Offline Policy Optimization, NeurIPS 2020
[3] MOReL: Model-Based Offline Reinforcement Learning, NeurIPS 2020
[4] An Optimistic Perspective on Offline Reinforcement Learning, ICML2020

In page 5, line 161, the epsilon in "parameter \epsilon" should be bold,  \boldmath{\epsilon}

Please check the bibliography for accidental lower case letters, like „q-learning“

Please check consistency: In [14] it’s „Hado Hasselt“, in [16] and [38] it’s „Hado Van Hasselt“, in [39] it’s „Hado van Hasselt“, which I believe to be correct.

**After rebuttal** \
I have read the other reviews and the feedback from the authors and think that the authors have responded well to the suggestions. I raised the score to 8.


**Time Spent Reviewing:**

7

---

> ### Author Response · Authors · 2021-08-10
> **Response to Reviewer cED7**
>
> Thank you for your very positive assessment of the significance, quality & clarity, and originality of the paper, and for your comments, which we will make sure to work into a revised version of the paper. In particular, thank you for suggesting the additional references that indeed will help contextualizing our work.
>
> We particularly appreciate you pointing out the value of the tandem RL setup as an analysis tool - this was in fact the original motivation for its development, for studying questions about the learning dynamics of RL algorithms in separation from the impact on behaviour, exploration and data generation. Due to length limitations we could not expand on this aspect, but we hope that our brief remarks on this in Section 4 (with an example application in the Appendix) will point readers in this direction and inspire future use of this tool beyond the scope of offline RL.

---

### Official Review · Reviewer_2Fyh · 2021-07-15

**Rating:** 7
**Confidence:** 4

**Summary:**

This paper studies the situation of passive reinforcement learning, simulating the famous Held and Hein twin cats paper where an active agent collects data to learn while the passive agent experiences the same data (but does not choose actions). The result, running Double-DQN on Atari, is similar - that the passive learner is often much worse. The paper then attempts to study why by evaluating three hypotheses: bootstrapping, data distribution, and function approximation, through a sequence of experiments. The conclusion is mixed, but mostly supports that data distribution is the key problem, with good evidence that bootstrapping is relatively less important than previously emphasized. Overall, the paper is a clever and thorough study of passive learning beyond the existing literature on offline RL.

**Limitations And Societal Impact:**

Generally, yes. However, the final broader impact paragraph as-is does not add too much value, you could use it to discuss limitations or consequences of understanding the important of active vs. passive in different application domains.

**Main Review:**

The main idea of the paper is to study “tandem” reinforcement learning: one agent (“active”) runs online RL as usual, another agent (“passive”) trains on the data of the active agent. Thus, the learning for the passive agent is different than offline RL because it is continuously receiving new data, but that data is not collected based on the agent’s own policy. Across Atari tasks, the passive agent performs significantly worse, unable to solve 3/4 of them (Fig 1 - all Atari envs are further included in the appendix and seem to have the same trend). This suggests similar issues as offline RL also exists in passive learning - it is not just additional data that is important but that the data comes from the agent’s own policy. The rest of the paper is devoted to studying why this happens and evaluating three hypotheses: (1) bootstrapping, (2) data distribution, and (3) function approximation as the major issue that causes the gap.

Section 3.1: this experiment was very interesting! For bootstrapping, the passive agent’s target is replaced by the active agent. The results are still poor for the passive agent, even when the passive agent is using the same exact targets as the active agent. This is quite surprising, as bootstrapping is often identified as one of the difficulties of off-policy RL (eg. the deadly triad). I believe it would be just as informative to perform the same study with the active agent, and I hope this can be included in the rebuttal.

Section 3.2: these sequence of experiments study data distribution. 1. The exploration parameter epsilon being higher reduces the gap between active and passive. This is somewhat unsurprising and the presentation of relative performance makes it a bit hard to understand if the issue is that active performance deteriorates or passive performance improves. 2. Sticky-actions does not change much; I did not really understand the point of this experiment and it seems like it could left out or moved to the appendix. 3. Larger replay buffers improve the passive agent and close the gap substantially to active. One question is whether it also improves the active agent. This is quite interesting though, suggesting that if you are limited to a small buffer, then the data distribution of that buffer being drawn from your own actions is very important. But perhaps with enough data it matters less (this would be consistent with standard deep learning). 4. If the passive agent is forked from an active agent in the middle of training, the performance deteriorates. This is also interesting and surprising, that the wrong data distribution can actually cause you to forget how to solve the task (even though the new data is solving the task!). One thought here is that perhaps bootstrapping is the culprit, and experiments similar to those in section 3.1 might disentangle why this deterioration happens - although this is partially addressed by the “on-policy evaluation” experiment. 5. Instead of freezing the policy, the replay buffer is frozen; there is still deterioration but slightly slower - again surprising. It is especially surprising this works better than the previous experiment, because I would have thought overfitting is a major issue with fixed replay buffer. 6. For forked tandem, using SARSA instead of Q-learning does not solve the deterioration, so ood bootstrapping may not be the culprit. 7. Adding data generated from the passive agent closes the gap, and closes it relatively quickly from 10% making a sizable difference to 50% closing the entire gap.

Section 3.3: Finally, function approximation is evaluated as the issue. First, ADAM and RMSProp is compared; no major difference. Adding more gradient updates per step makes the passive agent worse. Larger width networks seem to close the passive-active gap somewhat, but larger depth networks do the opposite. These findings are interesting but basically inconclusive. In Figure 8, a varying amount of the first k layers is shared between the active and passive agent, and the results are that sharing more layers helps, with sharing 3/5 closing about half of the performance gap and sharing 4/5 closing the entire performance gap (at that point it is linear so if the active one works, the passive one should too). This result, that sharing the early layers improves performance this much, is also surprising to me - the lower layers being trained on the same objective but better/different data significantly changes the outcome (although perhaps this is due to bootstrapping in the end?).

There is no related work; the paper instead covers prior work in the intro and throughout. This could be okay, but it is missing discussion of some modern offline RL work that studies similar problems:
[1] Conservative Q-Learning for Offline Reinforcement Learning. Kumar 2020
[2] Deployment-Efficient Reinforcement Learning via Model-Based Offline Optimization. Matsushima 2020.
[3] AWAC: Accelerating Online Reinforcement Learning with Offline Datasets. Nair 2020.
[4] Behavior Regularized Offline Reinforcement Learning. Wu 2019.
[5] Model-based Offline Policy Optimization. Yu 2020.

The concluding conjecture - deep RL is unstable on any fixed data distribution - seems too strong, and then vacuous when hedged with - “without explicit regularization or additional inductive bias.” All of offline RL is basically trying to find the right balance of expressivity and regularization, and while this paper provides interesting experiments in this direction towards the data distribution depending on the current policy being important, it doesn’t preclude some form of regularization that alleviates the issue.

Generally the paper is well organized and very clear and enjoyable to read.

Minor comments

Maybe I missed - in forked tandem, where does the stochasticity come from? Is it from sampling in epsilon greedy exploration, environment stochasticity, or something else?

It is considered bad style to use a citation as a noun, eg. line 175 “offline results in [3]” instead of writing a method or author name - it makes it harder to read.

**Time Spent Reviewing:**

4

---

> ### Author Response · Authors · 2021-08-10
> **Response to Reviewer 2Fyh**
>
> Thank you for your positive assessment of the clarity and relevance of the paper, and the great detailed feedback, suggested references, questions & comments. We attempt to address these one-by-one below:
>
> **Sec 3.1 & performing the bootstrap study with the active agent**: The focus of our paper was to study the learning process of the passive agent, given data provided by the active agent. Given that the active agent is a standard RL agent (that receives no signal from the passive agent), it is not clear to us what insight the reviewer has in mind when suggesting changing the active agent's target?
>
> **Sec 3.2, 1. (epsilon)**: We agree that the 'relative performance' reporting can obscure the effect of a parameter variation on the active performance itself, so we included absolute performance plots in the appendix (Figure 15). Because of length limitations we were unable to fit both in the main text, and felt that the 'relative' plots are more compact and informative given that our primary focus is on the magnitude of the 'tandem gap' rather than the absolute active performance. For these particular experiments the effect of the epsilon parameter on active performance in the chosen games is relatively small, and so relative plots are in fact mostly comparable to the absolute ones (Fig 3 vs Fig 15). We will add a clarifying remark to this effect in the main text.
>
> **2. (sticky actions)**: This experiment's motivation was to address the question whether data diversity (in different senses, e.g. caused by various sources of stochasticity) can be linked to the magnitude of the tandem effect. While both epsilon and sticky actions cause certain randomness and hence diversification in the data, sticky actions do not actually increase diversity in action space, while higher epsilon values do - the resultant impact on the 'tandem gap' seems to confirm our intuition that diversity & coverage in action space are more critical than other stochasticity in the transitions. We agree this is among the less informative experiments and will consider moving it to the appendix.
>
> **3. (replay size)**: Indeed here we varied the replay size for the *passive agent only* (hence the absolute performance plots, Fig 4, and the caption "for passive agent") - while the active/passive agents share the same replay buffer, the active agent only ever samples from the latest 1M transitions as is the default, while the passive one samples uniformly from the last 1M, 2M or 4M transitions. We will clarify this in the main text.
>
> **5. (forked tandem, fixed replay vs fixed policy; also re: "with more data, own data is less important")**: Our intuition here is indeed one of 'over-fitting'. Note that while nominally, the fixed-policy setting produces a 'larger' dataset  - a virtually infinite stream of data from a fixed non-deterministic policy - the 'relevant diversity' of this dataset may very well be smaller compared to a fixed-size dataset produced by a sequence of different policies (as in the fixed replay setting). So while in principle we agree that the amount of diverse data can reduce the dependence on own data, it is likely that the exact nature of this "data diversity" plays a significant role.
>
> **6. (forked tandem)**: We would like to add that MC-return regression policy evaluation (mentioned in the main text and expanded on in appendix A.3) may be making the conclusion of bootstrapping not being the main culprit even more strongly than the SARSA variant.
>
> **Additional references**: Thank you for providing the list of additional references - we will work these into our main text, as these fit very well with the context in which we see our work.
>
> **Regarding the conjecture**: We agree that this claim is somewhat imprecise, contentious without the caveat of some regularization potentially helping (as it does in many of the practical offline RL successes!), and perhaps weak as a formal claim once that caveat is added. At the same time we believe that the underlying intuition (that deep RL training may typically be inherently unstable on *any fixed* data distribution, independently of the common on/off-policy distinctions) may point in an important and under-appreciated direction, and put the role of interactivity into a sharper focus. Whether the conjecture itself is true or false (theoretical results in either direction would be highly interesting!), we would hope for it to point to important questions and trigger interesting discussion. We will attempt to find a better formulation but would try to retain the core intuition gained from the tandem work and expressed here.
>
> **Regarding the source of stochasticity in the forked tandem**: The main sources of stochasticity in the forked tandem experiments are eps-greedy exploration (of the active agent), uniformly-random sampling from the replay buffer (for the passive agent), and the eps-greedy behaviour policy at evaluation time (by the passive agent).
>
> **Regarding the comment on the use of the limitations & societal impact section**: Thank you for the suggestion, we will add some discussion on these points.
>
> Thank you for the stylistic comment on reference style, we will fix this throughout the paper.

---

### Official Review · Reviewer_kz5A · 2021-07-20

**Rating:** 7
**Confidence:** 4

**Summary:**

This paper provides detailed empirical analysis to study the challenge of offline reinforcement learning. The authors propose to use the Tandem RL setup as the analytic tool, where one active agent performs usual online training loop, while the passive agent can only learn from the data generated by the action learner. The results provide some insights for revealing the difficulty of offline RL.


**Ethical Concerns:**

No ethical issue has been found.

**Limitations And Societal Impact:**

No societal issue has been found.

**Main Review:**

Contributions:

The paper is very well written. The definitions and experiment setups are clearly defined. I really enjoy reading this paper.

The main message of this paper is that, as function approximators for value functions, deep neural networks are extrapolating in an uncontrolled fashion especially under insufficient coverage of the fixed data set, and as such blow up errors in an iterative training scenario. I think this is well expected as even in the linear case such extrapolation errors occur as suggested by previous theoretical works.

It is really good to see that the authors actually try to provide detailed analysis to reveal the hardness of offline RL. Most previous works in deep offline RL literatures just try to demonstrate their proposed algorithms are good in some cases, but it’s rarely to see offline RL algorithm can fail very easily, and what are the major factors that cause the trouble.

Weakness:

I have the following comments and questions that I hope the authors to address:

What does the over-generalization exactly mean in the paper? Does that mean over estimate the unseen data or something even stronger? Is it possible to visualize such over-generalization?

I think the data distribution assumption (Hypothesis 2) as well as the supporting experiments can be strengthened such that it can distinguish between "enough data coverage"  and "enough on-current-policy data coverage". It has been shown that the "enough data coverage" is not enough to guarantee efficient offline RL (e.g. see [2,3]). Maybe the experiments shown in the paper connects the failure of passive learning to the later one?

Almost all experiments are in the policy optimization problem. I am wondering if the hardness of offline RL already exists in the simpler policy evaluation problem since the Hypothesis 2 and 3 seem also exist there (although I understand it's hard to design experiments for policy evaluation in this tandem setups).

There is also some recent theoretical paper [1] trying to reveal the difficulty of offline RL. But I guess the major factor there is the estimation error that comes from the insufficient coverage of the data (as they consider the tabular setting so there is no extrapolation error while the problem still exists). I am wondering if the authors distinguished the estimation error and extrapolation error in this paper?

[1]  Xiao, C., Lee, I., Dai, B., Schuurmans, D. and Szepesvari, C., 2021. On the Sample Complexity of Batch Reinforcement Learning with Policy-Induced Data. arXiv preprint arXiv:2106.09973.

[2] Wang, R., Foster, D.P. and Kakade, S.M., 2020. What are the Statistical Limits of Offline RL with Linear Function Approximation?. arXiv preprint arXiv:2010.11895.

[3] Wang, R., Wu, Y., Salakhutdinov, R. and Kakade, S.M., 2021. Instabilities of offline rl with pre-trained neural representation. arXiv preprint arXiv:2103.04947.

**Time Spent Reviewing:**

3

---

> ### Author Response · Authors · 2021-08-10
> **Response to Reviewer kz5A**
>
> Thank you for the positive assessment of the writing and the relevance of our analysis, as well as the very useful feedback, the references and great questions. We will work in the references and your comments in a revised version of the paper. We attempt to address your comments & questions below:
>
> **Regarding "over-generalization"**: The term 'over-generalization' in the paper is meant in the sense of a function approximator's tendency to wrongly assign similar values from seen states to unseen ones (akin to underfitting). Other forms of 'extrapolation error' (also touched on) could be, e.g. an effect like overfitting, whereby overly precise fitting on seen data leads to degenerate and unrealistic extrapolations on unseen data. We will attempt to make the distinction clearer in the text - a more detailed analysis however, e.g. via measuring gradient interference, seemed beyond the scope of our work. While 'over-generalization' seems hard to capture quantitatively in our setting, anecdotally we have observed extrapolation error in the form of 'gradual value explosions' (a tendency of unseen action-values to creep upwards over the course of passive training, often reaching completely unrealistic value ranges) in some, but not all, of our experiments. We hypothesize that this effect, while forming only part of the 'tandem effect', is in fact the part most driven by bootstrapping. We will try to obtain and include a more quantitative confirmation of this hypothesis for the final version of the paper.
> This is also connected to your question about extrapolation vs estimation error: Indeed both pose a challenge for offline RL, and insufficient data coverage can of course even affect the tabular setting via estimation error. On the other hand, the focus of our notion of the  'tandem effect' is on the specific challenge of erroneous extrapolation & over-generalization in the function-approximation case. Here, exposure to data is a priori demonstrably sufficient to overcome estimation error - via the active agent's success to learn (also considering the experiment in Appendix Fig 19 of more updates worsening passive performance). In this case, over-generalization or other erroneous extrapolation lead to passive learning failure nonetheless.
>
> **Regarding "data distribution assumption"**: We agree with the reviewer's intuition, and the proposed distinction runs along very similar lines to our thinking. Our experiments do not yet allow to conclusively draw the line between the two types of data coverage, and a more thorough analysis of this detail is left to future work. On the other hand, it is true that some of our results provide preliminary support to this distinction - e.g. epsilon/replay-size experiments (Figs 3 & 4)  can be interpreted as mostly modifying the 'general data coverage', while the own-data experiment (Fig 6) probes the effect of 'on-current-policy data coverage' and the forked-tandem experiments sit somewhere in between. We will try to highlight this intuition in the paper without making definitive claims.
>
> **Regarding policy evaluation**: The question of whether the Tandem Effect applies to policy evaluation (rather than just control) is somewhat subtle: On the one hand, it indeed does occur in pure policy evaluation, as our experiments confirm - the penultimate paragraph of Section 3.2 mentions our SARSA and MC-return regression experiments, performing policy evaluation in the forked tandem setting (further results for this scenario are included in the Appendix (A.3)). On the other hand, the Tandem Effect is a 'failure to control' - while policy evaluation itself does not fail in the above scenarios (on-policy values are learned accurately), the acquired value function does not admit adequate control because of basically unconstrained values occurring in state-action pairs atypical under the given data distribution, but potentially relevant to control performance. We will attempt to highlight our policy evaluation results more clearly in the main text, as we believe these strengthen the intuition for the effect and once more demonstrate the exacerbating but not causal role of bootstrapping (Hyp 1) in this context.

---

### Author Response · Authors · 2021-08-26
**Checking in regarding remaining questions**

We have noted that none of the reviewers have engaged further with the discussion. While we take this as a positive sign that most or all of their concerns were alleviated, we would like to check in if there are any concerns we have not addressed, before the discussion period is over - especially if addressing those would potentially affect any of the given scores.

---

### Decision · Program_Chairs · 2021-09-27

**Decision:**

Accept (Poster)

**Comment:**

The paper analyzes the reason for the poor performance of learning control policies from offline datasets. The main conclusion is that
- Insufficient coverage of suboptimal actions to poor performance when the policy is evaluated is online.
- This problem is further aggravated by deep networks that extrapolate value function in the regions of the space that are infrequently explored by the agent.

Three reviewers vote for accepting the paper, while reviewer LigT dissents. The concerns about (i) more seeds; (ii) increased mathematical rigor will definitely make the claims of the paper stronger. However, they are not a ground for rejection. I encourage the authors to incorporate these suggestions.